# Integrating causal inference and machine learning to quantify climate-malaria relationships: Evidence of temperature and rainfall thresholds from Colombian municipalities

**Juan David Gutiérrez** [ID] *

Facultad de Ciencias Médicas y de la Salud, Universidad de Santander, Instituto Masira, Bucaramanga, Santander, Colombia

* jdgutierrez@udes.edu.co

## Abstract

Rainfall and temperature are key climate determinants of malaria incidence; yet their causal exposure-response curves on malaria incidence across the entire Colombian territory remain unquantified. We estimated the effects of rainfall and temperature on malaria incidence at the municipal scale from 2007 to 2023. We conducted an ecological observational study in 969 Colombian municipalities located below 1,600 meters. The monthly Standardized Incidence Ratio (SIR) of malaria was calculated for each municipality. Directed acyclic graphs guided the identification of the appropriate adjustments needed to emulate the corresponding experimental design and avoid inducing bias, and the effect was estimated using a modified approach of the Targeted Maximum Likelihood Estimation (TMLE). Exposure-response curves were estimated for two outcomes: the current month and the moving average for the current and previous month. A total of 1,075,112 cases of malaria were reported. The results suggest a non-linear relationship between rainfall and temperature concerning the SIR of malaria, indicating an optimal temperature of 25 °C and approximately 37 mm of rainfall for the highest incidence. The negative control test revealed the presence of residual confounding bias (p < 0.05) in all estimates. Meanwhile, the estimations of the E-value indicated low to moderate tolerance (E-value = 1.14 − 1.48) to an unmeasured confounder. These findings support the integration of rainfall and temperature thresholds into early-warning systems for targeted malaria control.

## 1 Introduction

Malaria is a life-threatening parasitic disease caused by protozoa of the *Plasmodium* genus, transmitted to humans through the bites of infected female *Anopheles* mosquitoes [1]. Globally, it remains a major public health challenge, with an estimated 249 million cases and 608,000 deaths reported in 2022, with the WHO African

**Data availability statement:** For replication purposes, the code and dataset are available at: https://github.com/juandavidgutier/malaria_exposure_response.

**Funding:** The author(s) received no specific funding for this work.

**Competing interests:** The authors have declared that no competing interests exist.

Region bearing the most significant burden [1]. In the Region of the Americas, progress towards malaria elimination has stalled, and the region reported approximately 600,000 cases in 2022, with Venezuela, Brazil, and Colombia accounting for 73% of all cases [2]. Nationally, the disease is endemic in over 80% of the territory. In 2023, the country reported 102,181 cases, with *Plasmodium falciparum* and *Plasmodium vivax* as the predominant circulating species, primarily affecting departments such as Chocó, Nariño, and Antioquia [3].

Malaria transmission is intrinsically linked to climatic and environmental conditions, which dictate the geographic distribution and intensity of the disease. Key climatic variables such as temperature, rainfall, and humidity are known to significantly influence the developmental and survival rates of both the *Anopheles* mosquito vector and the *Plasmodium* parasite within it [4,5]. Specifically, warmer temperatures can accelerate the sporogonic cycle of the parasite, while rainfall patterns often create the necessary aquatic habitats for mosquito larvae to thrive [6].

The relationship between climatic variables such as temperature and precipitation and the incidence of malaria has shown evidence suggesting the existence of threshold values for both variables that influence transmission dynamics. Below these thresholds, malaria incidence tends to increase as conditions become conducive for mosquito breeding and parasite development. For instance, mean monthly temperatures ranging from 16 °C to 25 °C have been associated with higher malaria incidence rates due to their favorable impact on vector survival and parasite development cycles [4,7]. Similarly, numerical simulations for South Africa suggest precipitation levels below 95 mm per month are linked to an increase in malaria transmission as they provide adequate breeding sites for mosquito vectors without excessive water that might disrupt larval habitats [4].

Conversely, exceeding these thresholds appears to result in a decline in malaria incidence. Temperatures above 25 °C are known to negatively affect mosquito survival, thereby reducing transmission potential [8]. Similarly, precipitation levels beyond 125 mm per month may lead to flooding, which destroys mosquito breeding sites and disrupts transmission cycles [4]. Understanding these threshold dynamics is crucial for predicting malaria outbreaks and developing effective control strategies [9].

Furthermore, anthropogenic environmental changes, including deforestation and altered land use for agriculture, can modify local hydrology and temperature, often increasing the suitability of habitats for malaria vectors and bringing human populations into closer contact with them [10]. Therefore, a thorough understanding of these complex interactions is crucial for developing robust causal models and effective public health strategies aimed at controlling and eliminating malaria in a changing world [11].

Previous studies have highlighted the significance of spatial and temporal clustering in malaria incidence, particularly in regions with shared environmental and socio-economic characteristics. Our 2023 research identified high-incidence malaria clusters in Colombia, predominantly in the Amazon, Pacific, and Urabá-Bajo Cauca-Alto Sinú regions, driven by factors such as forest coverage and rainfall patterns [12]. Building on these findings, this study extends the research question to

climate variables into a causal framework to estimate the exposure-response relationship of temperature and rainfall with malaria incidence.

Note that the gold standard for estimating the causal relationship between climate variables and malaria incidence at the municipal scale is a set of experimental designs. However, such experimental studies are not possible due to practical and ethical reasons [13]. In this scenario, the best option is likely to implement causal inference using observational data. In our research, we utilized observational data from an epidemiological source and remotely sensed climate variables, implementing causal machine learning to address the question of the relationship between temperature and rainfall with malaria incidence in Colombian municipalities.

Understanding the intricate relationship between climate variables and malaria incidence is crucial for effective public health policy, particularly in regions like Colombia, where the disease is endemic. Changes in temperature and rainfall patterns significantly influence mosquito vector habitats and parasite development cycles, altering transmission dynamics [11]. This research seeks to provide evidence-based thresholds for rainfall and temperature that can inform Colombia's malaria elimination strategies, enabling targeted interventions and early-warning systems to mitigate outbreaks [8,14]. Our results can offer actionable insights to enhance climate-sensitive disease surveillance and control measures in vulnerable regions. This provides feasible evidence for decision-makers and public health authorities, who require high-quality information to design new malaria control programs or reformulate current public health agendas for vector-borne diseases.

The study aims to causally assess how varying levels of monthly rainfall and temperature influence malaria incidence in Colombian municipalities during the period from 2007 to 2023. The focus is on estimating the shape of the exposure-response relationship nonparametrically by implementing a causal machine learning approach. The findings from this study are designed to enhance climate monitoring initiatives and contribute to improving intervention approaches for malaria eradication in areas with heightened vulnerability across the country.

## 2 Methods

An ecological observational analysis was performed utilizing openly available, anonymized case records across Colombian municipalities (i.e., the observation units were the municipalities). Ethical authorization for this scientific inquiry was granted by the Bioethics Committee at Universidad de Santander (reference code 002, dated February 13th, 2023). The study follows STROBE reporting standards [15], international protocols designed to improve methodological transparency in health research.

### 2.1 Epidemiological data

On September 14, 2024, epidemiological records detailing confirmed malaria cases (as determined by laboratory tests) across Colombian municipalities were obtained from SIVIGILA, the national health surveillance platform, for all forms of malaria. The compiled dataset spanned 17 years of observations (January 2007 – December 2023). For analytical rigor, entries with discrepancies in occurrence location, demographic details, or date inconsistencies were systematically excluded from the assessment. Furthermore, data from municipalities exceeding 1,600 meters in elevation were excluded, based on the suggestion of Rodríguez et al. (2011) [16]. This decision aligns with altitude-dependent epidemiological models where cooler temperatures at higher altitudes create unsuitable conditions for malaria vectors [17].

To assess the exposure-response curves of rainfall and temperature on malaria transmission patterns, our analysis employed monthly calculations of the Standardized Incidence Ratio (SIR). This metric compares observed infection counts in municipalities against projected baseline figures derived from nationwide demographic estimates provided by the National Statistics Department [18]. The estimation of monthly SIR across municipalities was performed utilizing the epitools R package (v0.5-10.1) [19]. Demographic adjustments were implemented through the indirect standardization method [20], aligning with the WHO's age groups [21] to ensure comparable incidence rate calculations across population groups.

## 2.2 Environmental data

**2.2.1 Hydro-climate variables.** We collected monthly satellite-sensed hydro-climate data of rainfall, air temperature at 2 m, and soil moisture up to 7 cm, from January 2007 to December 2023, from the ERA5 dataset [22], with a spatial resolution of 0.10 degrees. We used the raster package in R (version 4.0.3) [23] to spatially match and estimate the monthly average of each hydro-climate variable for each municipality.

**2.2.2 Sea Surface Temperature.** We retrieved seven monthly indices of sea surface temperature (El Niño region 1–2, El Niño region 3, El Niño region 3–4, El Niño region 4, north Atlantic (5–20°North, 60–30°West), south Atlantic (0–20°South, 30°West-10°East), and global Tropics (10°South-10°North, 0–360)) from the National Oceanic and Atmospheric Administration (NOAA) database, covering the period from January 2007 to December 2023 [24]. The indices of sea surface temperature were included in the analysis as potential confounding factors because of their simultaneous association with rainfall and temperature patterns in continental regions [25], and their relationship with the incidence of malaria, modifying other hydro-climate variables associated with the occurrence of the disease, such as relative humidity, runoff, and soil temperature [14,26,27].

**2.2.3 Vegetation variables.** We obtained annual raster layers for the period 2007 – 2023 of forest coverage from the NASA product MCD12Q1 [28], with a spatial resolution of 500 m, and monthly raster layers for the same period of the Enhanced Vegetation Index (EVI) from the NASA product MOD13A3.061 [29], with a spatial resolution of 1 Km. The spatial matching between vegetation variables and municipality polygons was developed as mentioned above. We reported the annual percentage of area in each municipality with forest coverage and the monthly average EVI for each municipality.

**2.2.4 Vector co-occurrence.** Eight *Anopheles* species of medical importance in Colombia (*An. albimanus*, *An. darlingi*, *An. nuneztovari*, *An. calderoni*, *An. oswaldoi*, *An. pseudopunctipennis*, *An. punctimacula*, and *An. rangeli*) were analyzed through environmental niche modeling. Occurrence records were obtained through the Global Biodiversity Information Facility [30] (S1 Fig), with spatial filtering applied to eliminate clustering effects. This involved retaining one randomly selected observation per 25 $km^2$ grid cell across 100 iterations using the spThin R package [31], generating species-specific georeferenced databases.

Species distribution modeling was conducted using the MaxEnt algorithm within the kuenm R framework [32]. The analysis partitioned data into training (80%) and validation (20%) subsets, incorporating 19 bioclimatic predictors from World-clim [33]. Model configurations included linear, quadratic, and multiplicative feature types. Optimal models were identified through multi-criteria evaluation: statistical robustness (partial ROC-AUC), predictive accuracy (omission error rates from test data), and parsimony (AICc-adjusted metrics). Spatial validation protocols (i.e., cross-validation) and 10,000 background points ensured performance assessment.

Model outputs were thresholded using the minimum training presence criterion (5% permissible omission error) to generate binary suitability layers [34]. These layers were integrated with the raster R package to quantify overlapping distributions. Municipal-level analyses identified regions where four or more vector species exhibited co-occurrence, expressed as the percentage of total municipal area.

## 2.3 El Niño cycle episodes

Classification of La Niña, El Niño, and Neutral conditions was based on standards from four climate agencies: the NOAA [24], the Meteorological Office of the Australian Government (MOAG) [35], the Tokyo Climate Center (TCC) [36], and the Colombian Institute of Hydrology, Meteorology, and Environmental Studies (IDEAM) [37]. Since these institutions use varying classification methodologies, we employed a conservative approach demanding concordance across all four agencies' standards to ensure analytical consistency and enhance reliability (S1 Table). The numerical coding system applied was: 0 for Neutral periods, 1 for La Niña occurrences, 2 for El Niño occurrences, and 3 for instances where institutional consensus was absent.

## 2.4 Socioeconomic data

In 2018, during the national census, the Multidimensional Poverty Index (MPI) was measured in the whole Colombian territory. MPI represents a socioeconomic metric that quantifies the proportion of families experiencing multi-faceted poverty within individual municipalities across the country. This comprehensive index incorporates six key areas: educational accessibility, conditions affecting children and young people, employment prospects, healthcare availability, access to public services, and residential standards. The MPI information was obtained from the National Department of Statistics [38].

## 2.5 Causal inference analysis

The aim of this study was to estimate how changes in rainfall or temperature would causally influence malaria incidence across Colombian municipalities using observational data. We emulated a hypothetical experiment through a causal framework that specifies how variables are related and which factors must be controlled to avoid bias. We used Directed Acyclic Graphs (DAGs) to represent our assumptions about the relationships between climate variables, socioeconomic and environmental factors, and malaria incidence (see S1 Text).

DAGs are graphical tools that make explicit which variables may act as confounders—that is, factors that influence both the exposure (rainfall or temperature) and the outcome (malaria incidence)—and which variables could induce bias if adjusted for, such as colliders or mediators [39]. This approach allowed us to decide, systematically, which covariates to include in the adjustment set and which to exclude to avoid bias amplification.

## 2.6 Machine learning implementation

To estimate the causal relationship between climate variables and malaria incidence, we combined traditional causal inference principles with flexible predictive models from machine learning. The analysis proceeded by first preprocessing the data to improve model stability and interpretability. Temporal variables were transformed to capture the seasonal cycle, exposures were restricted to biologically meaningful ranges (15–30 °C for temperature and 0.01–60 mm for rainfall), and missing data were handled through complete case analysis. These steps ensured comparability across municipalities and prevented extrapolation to unusual climatic conditions.

We first modeled how malaria incidence changes with rainfall and temperature after accounting for other factors (the outcome model) and simultaneously modeled how rainfall and temperature vary across municipalities given those same factors (the exposure model) [40]. The final causal effect was obtained by combining both predictions in a way that corrects for residual confounding and model misspecification. This approach, known as doubly robust estimation, yields consistent results even if one of the two models is imperfect. Machine learning methods, specifically gradient boosting algorithms, were used to flexibly capture complex, nonlinear relationships between exposures and malaria incidence without assuming a predetermined shape.

The resulting exposure–response curves represent the expected change in malaria incidence under different levels of rainfall and temperature, together with uncertainty intervals that reflect sampling variability. Although the estimation relies on machine learning, the interpretation remains epidemiological: we estimate what would happen, on average, to malaria incidence if the climatic conditions were to change, assuming the identified confounders have been adequately controlled.

Furthermore, we estimated the exposure-response curve for rainfall and temperature on the SIR of malaria for the current month, as well as the moving average of the current month and the previous month.

## 2.7 Robustness and sensitivity tests

Because observational studies are always susceptible to hidden biases, we implemented a series of diagnostic tests to evaluate how robust our causal findings are to potential unmeasured confounding. The first test was a negative control exposure designed to detect residual bias. A negative control is a variable that cannot have caused the outcome but may

share the same unmeasured causes as the exposure. In our case, we used future values of rainfall and temperature (for the subsequent month) as negative controls, based on the logical principle that future climate conditions cannot influence past malaria cases. If those future values show a statistically significant association with past malaria incidence, it indicates that some unmeasured factor may still be confounding the estimated effect [41].

The presence of a significant association ($p < 0.05$) suggested residual confounding, meaning that while the overall pattern of the exposure–response relationship remains credible, the absolute magnitude of the estimated effect should be interpreted with caution.

In addition, we implemented the E-value sensitivity analysis, a quantitative tool that expresses how strong an unmeasured confounder would need to be to completely explain away the observed effect. An E-value close to one indicates that even a weak confounder could alter the result, while larger E-values imply that only an implausibly strong confounder could do so. This measure does not eliminate uncertainty but contextualizes it, helping to judge whether the detected effects are likely to be robust in the presence of unmeasured confounders.

Technical details and the complete statistical specification of the causal inference, machine learning, and refutation tests are provided in the S1 Text, for replication and further scrutiny.

## 3 Results

In the 969 municipalities included in the study, a total of 1,075,112 cases were reported between 2007 and 2023. Most cases occurred in males (n = 713,874; 66.4%). The years with the highest number of reported cases were 2010 and 2023, with 99,745 and 96,103 cases, respectively. The municipalities with the highest average SIR between 2007 and 2023 were mainly located in the western, eastern, and southern regions of the country (Fig 1).

During the study period, northern and eastern regions of the country had the highest monthly average temperature, with monthly values above 25 °C; meanwhile, most of the Andean region showed temperatures below 17 °C (Fig 2a). The Western region exhibited the highest rainfall values in the country, with monthly values above 55 mm (Fig 2b) between 2007 and 2023.

The R software ggdag allows us to identify the appropriate adjustments for estimating the effects of rainfall and temperature on the SIR of malaria. To preserve causal validity, we identified variables that act as colliders through our DAG analysis. In causal inference, a collider is a variable that is influenced by multiple variables simultaneously. Note that when such variables are adjusted for in the analysis, they create a spurious association between the exposure and outcome—a phenomenon known as collider bias.

For the rainfall effect estimation, our DAG revealed that soil moisture, temperature, vectors, EVI, and MPI are colliders because they receive arrows from multiple paths, including both climate variables and malaria-related factors (S1 Text). For example, in Colombia at the municipal scale, MPI is influenced by climatic conditions in the sense that municipalities in regions with extreme climate conditions tend to have larger MPI [16], while simultaneously being affected by forest coverage through the reduction of infrastructure and economic productivity, which leads to a reduction on social proxies involved in the measure of MPI, including healthcare access. Similarly, for the temperature effect, vectors, EVI, and MPI function as colliders.

As mentioned above, adjusting for these collider variables would have conditioned on a common consequence, artificially inducing an association between exposure climate variables and malaria incidence that does not reflect the true causal relationship of interest. Therefore, we omitted these variables from our final models to avoid introducing collider bias, ensuring our estimates represent unbiased causal effects rather than confounded associations [39,42–44]. This approach aligns with best practices in causal inference, where the structure of relationships, as depicted in the DAG, guides variable selection rather than merely considering statistical associations.

The estimation of the exposure-response curve for rainfall in the range of 0.01 to 60 mm for the current month showed a sharp threshold around 37 mm. Monthly rainfall values below this threshold favored the incidence of malaria in the

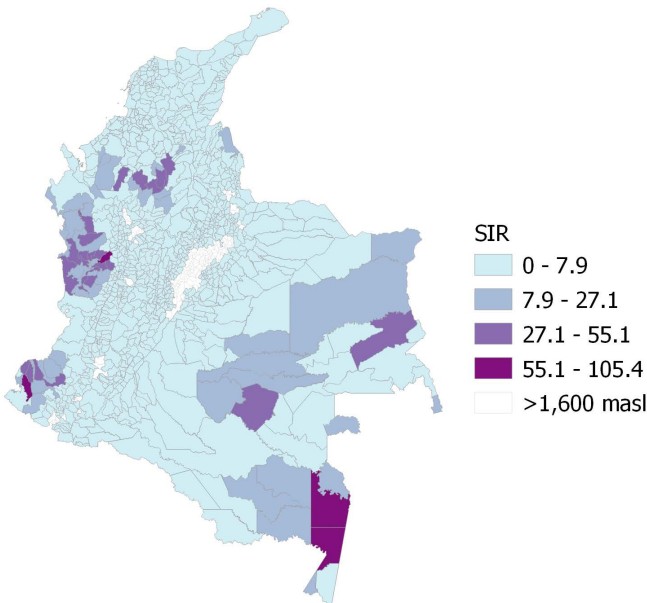

**Fig 1. Average standardized incidence ratio (SIR) for malaria from 2007 to 2023; masl = meters above sea level.** The map was created using QGIS software, with the basemap shapefile (https://www.dane.gov.co/files/geoportal-provisional/SHP_MGN2021_COLOMBIA.zip) sourced from the Colombian National Geostatistical Framework, an openly available resource (https://www.dane.gov.co/files/geoportal-provisional). The terms of use for the base shapefile are compatible with CC-BY 4.0 (https://geoportal.dane.gov.co/acerca-del-geoportal/licencia-y-condiciones-de-uso/#gsc.tab=0).

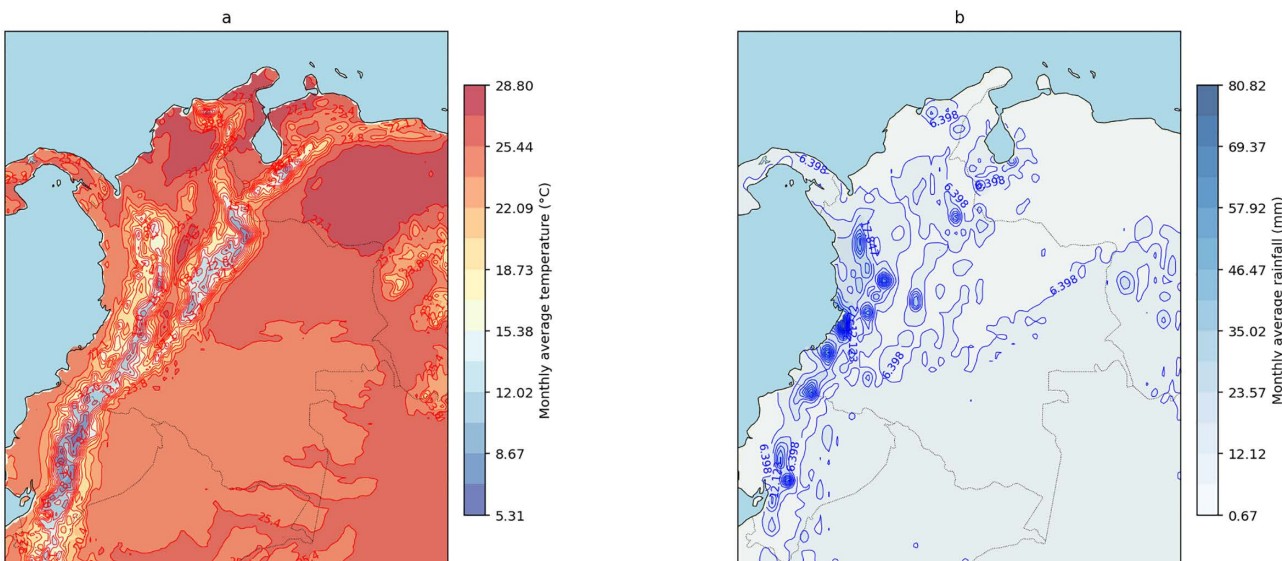

**Fig 2. Isomaps of average monthly temperature (a) and rainfall (b) for the Colombian territory, during the period 2007 – 2023.** The maps were created using Python software, with the basemap shapefiles (https://www.naturalearthdata.com/ http://www.naturalearthdata.com/download/110m/physical/ne_110m_land.zip) sourced from Natural Earth, an openly available resource (https://www.naturalearthdata.com/). The terms of use for the base shapefile are compatible with public domain standards (https://www.naturalearthdata.com/about/terms-of-use/).

municipalities, while values above 37 mm resulted in a reduction of malaria cases ([Fig 3a]). For the moving average of the current and previous month, the exposure-response curve showed a similar pattern but with a flat threshold of rainfall in the range of 34–42 mm ([Fig 3b]).

In the case of the exposure-response curve for temperature evaluated in the range of 15–30 °C, the shape of the curve was the same for the current month and for the moving average of the current and previous month ([Fig 4a] and [4b]). A sharp peak was observed at 25 °C. Temperature values below 25 °C increased the SIR in the municipalities. In contrast, values exceeding this threshold led to a decrease in malaria incidence. In both cases, temperatures above 28 °C appear to be associated with SIR values that are not significantly different from zero, according to the 95% confidence interval.

The application of the refutation test, which involves adding a negative control exposure, to assess residual confounding in our estimate of the effect of rainfall and temperature on the SIR of malaria, indicated the presence of residual confounding bias. This is evidenced by a statistically significant association (p-value < 0.05) between the future indicators (rainfall in t + 1 and temperature in t + 1) and the past SIR of malaria across all estimations.

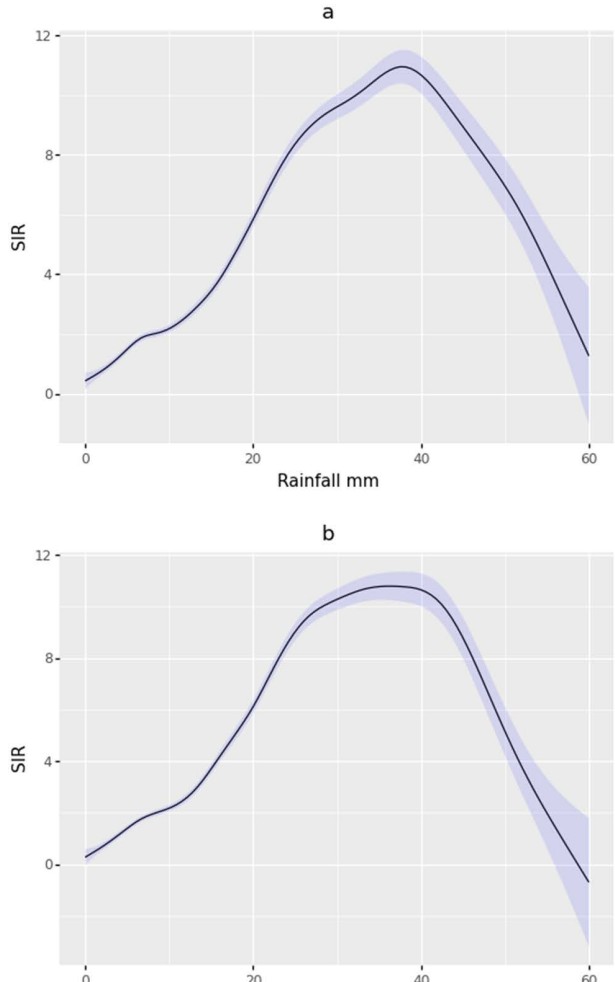

**Fig 3. Exposure-response curve for the effect of rainfall on the SIR of malaria in the current month (a), and for the moving average for the current month and the previous month (b).** The black line represents the response values of the SIR to the exposure (rainfall), and the blue band represents the 95% confidence interval.

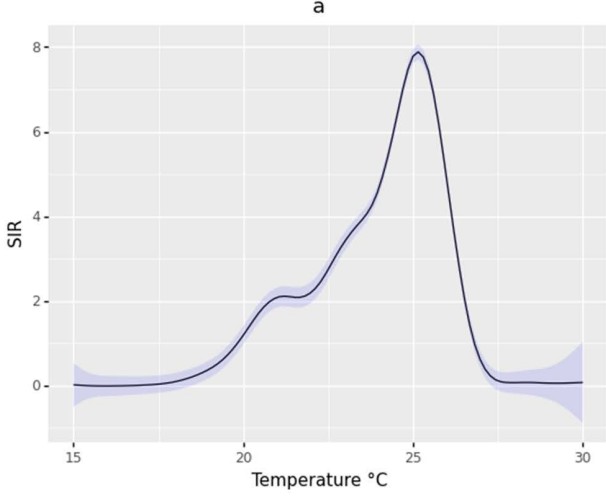

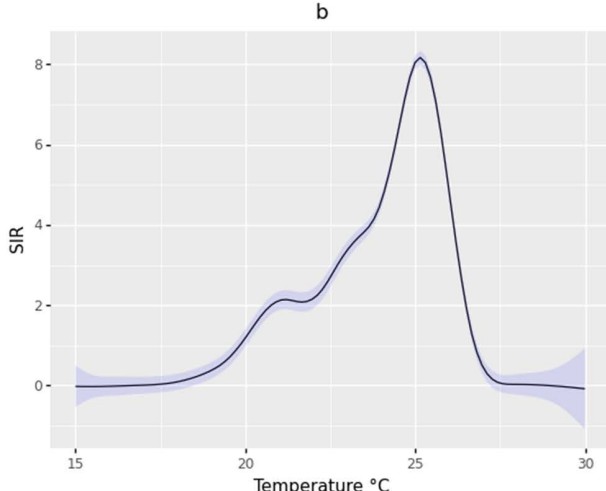

**Fig 4. Exposure-response curve for the effect of temperature on the SIR of malaria in the current month (a), and for the moving average for the current month and the previous month (b).** The black line represents the response values of the SIR to the exposure (temperature), and the blue band represents the 95% confidence interval.

The sensitivity test for potential unmeasured confounding showed E-values for the effects of rainfall on the SIR of malaria of 1.14 for the current month and 1.48 for the moving average of the current and previous month. For the effect of temperature, the E-values were 1.46 for the current month and 1.31 for the moving average of the current and previous month, respectively. These results indicate that some of our estimates are low-tolerant to the presence of an unmeasured confounder (e.g., the estimate for rainfall in the current month with an E-value = 1.14), while others are moderately tolerant (e.g., the estimate of the effect of rainfall for the moving average of the current and previous month with an E-value = 1.48).

## 4 Discussion

In this study, we estimated the exposure-response curve for the effects of rainfall and temperature on the SIR of malaria in Colombian municipalities with altitudinal suitability for malaria transmission during the period from 2007 to 2023. We

employed a causal machine learning approach to obtain estimates for the current month as well as for the moving average of the current and previous month. Our results confirm the existence of a non-linear relationship with a threshold value for the effects of rainfall and temperature on the SIR of malaria, approached from a perspective at the municipality scale and using observational data.

## 4.1 Temperature and rainfall gradients

The threshold value of 25 °C for the effect of temperature observed in this study is consistent with previous work by Mordecai et al. from laboratory studies and thermal biology of the mosquito [7,8]. Previous studies on a threshold for rainfall are much scarcer, and our results suggest that, for the Colombian territory, the threshold value is considerably lower than the 95–125 mm reported for the province of KwaZulu-Natal in South Africa by Okuneye and Gumel [4]. In Colombia, monthly rainfall values exceeding ~37 mm likely result in the flushing of breeding sites for the vectors.

The effect of temperature on malaria transmission is related to the survival and reproductive dynamics of *Anopheles* mosquitoes and the development of *Plasmodium* parasites. Studies have demonstrated that temperatures within the range of 20–30 °C are optimal for both vector survival and parasite development, with deviations from this range significantly reducing transmission potential [45]. For instance, at higher temperatures exceeding 30 °C, mosquito survival rates decline due to increased metabolic stress and reduced longevity, as noted in experimental studies exploring the thermal limits of malaria transmission [46]. Conversely, temperatures below 20 °C slow the sporogonic cycle of *Plasmodium*, thereby reducing the likelihood of successful transmission [47].

Additionally, fluctuating warm and humid conditions have been shown to differentially affect mosquito physiology, altering their feeding behavior and vectorial capacity, which further emphasizes the complex interplay between temperature and malaria dynamics [7]. These findings underscore the importance of incorporating local temperature variability into predictive models to enhance malaria control strategies in regions like Colombia.

The relationship between rainfall and malaria transmission is related to the dynamics of mosquito breeding habitats. While moderate rainfall is necessary to create standing water for mosquito larvae, excessive rainfall can lead to significant habitat disruption. For instance, studies have shown that heavy rainfall events can result in high mortality rates among immature mosquito populations due to the flushing of breeding sites [48].

Previous research assessing malaria incidence across regions with varying rainfall patterns has found that high rainfall correlates with reduced malaria cases, likely due to the destruction of larval habitats and the dispersal of mosquito populations [49]. These findings suggest that the threshold for rainfall conducive to malaria transmission is highly context-specific. In Colombia, where dense forest coverage and steep terrains are prevalent, even moderate rainfall (~37 mm) may suffice to maintain breeding sites without triggering flushing events. This highlights the need for region-specific studies to refine rainfall thresholds and improve predictive models for malaria control strategies.

## 4.2 Methodological approach

This study represents a methodological difference from previous research that has largely focused on establishing associations between hydro-climatic variables and malaria incidence using traditional statistical models [26,50–52]. By implementing a causal machine learning framework, we move beyond mere correlation/association to formally estimate the causal effect and the exposure-response curve of rainfall and temperature on malaria incidence.

Our approach can be particularly useful in complex observational settings as it leverages machine learning to flexibly model the relationships between a high-dimensional set of covariates and both the exposure and the outcome, thereby reducing bias in the final effect estimate [53,54]. The explicit use of a DAG to guide the analysis further enhances the transparency and validity of our causal assumptions, a practice that strengthens the inferential basis of our findings compared to conventional regression analyses [55].

## 4.3 Limitations and challenges

It is important to emphasize that the uncertainty intervals presented in this paper quantify sampling variability of the estimated causal functions under the assumptions of causal inference (S1 Text). These intervals do not capture systematic bias arising from violations of causal assumptions — notably residual confounding or measurement error — and therefore should not be read as bounds that include all sources of uncertainty. Furthermore, our study includes the assumption of independent and identically distributed data in our causal machine learning framework, but our dataset can also be modeled using a hierarchical structure. This kind of data structure may lead to underestimation of standard errors and overconfidence in the precision of our estimated exposure-response curves [56].

For the reason mentioned above, we complemented the intervals with sensitivity diagnostics, particularly negative control exposures and E-values. The former tests for residual confounding signals that would not be reflected in the intervals, while the latter quantifies how strong an unmeasured confounder would need to be to explain away the observed association. Interpreting the 95% intervals causally therefore requires accepting the identifying assumptions; when diagnostics indicate potential violations, the intervals should be interpreted as characterizing precision under the model rather than definitive evidence of causal magnitude.

This study sets a benchmark for methodological transparency in environmental epidemiology by applying a formal battery of refutation strategies to interrogate the core causal claim. The negative control exposure test revealed statistically significant associations ($p < 0.05$) between future climatic exposures and past malaria incidence, indicating the presence of residual confounding bias.

The results of the negative control exposure test imply that, despite the extensive adjustment set, some unmeasured or imperfectly measured variables—such as local malaria control interventions, population mobility, or diagnostic capacity—may still partially explain the observed causal effects. Consequently, while our estimated exposure–response functions reflect climatic dependencies, the absolute magnitudes of causal effects should be interpreted with caution.

Rather than invalidating the causal framework, the detection of residual confounding underscores the limitations of observational inference in complex eco-epidemiological systems and highlights the value of triangulation approaches, such as integrating natural experiments, in future studies [41]. Thus, the negative control result does not undermine our main conclusion about threshold behavior in malaria–climate relationships, but rather qualifies it—signaling that small unmeasured confounders could slightly shift the estimated exposure-response curves without altering their overall non-linear shape or biological plausibility.

Our analysis, while including a comprehensive set of variables, reveals significant trends but may not fully capture the influence of several critical municipal-level factors that could distort the observed relationship. For instance, the implementation and concentration of directed vector control campaigns are not uniform across Colombia, and are often intensified in response to perceived high-risk periods, such as rainy seasons [16].

In this sense, it is plausible that the enhanced distribution of insecticide-treated nets or the implementation of indoor residual spraying in certain municipalities during periods of high rainfall could independently reduce malaria transmission, thereby intensifying the negative effect attributed to the flushing of breeding sites. Furthermore, differential access to and capacity of local health services represent a significant source of potential bias.

Similarly, barriers to diagnosis, including geographical isolation and socioeconomic factors, are well-documented in Colombia's endemic regions and can lead to substantial underreporting of cases [57]. This issue is particularly exacerbated during extreme climatic events, such as the La Niña phenomenon, which is known to cause widespread flooding and landslides in the country [58]. Such events can severely disrupt transportation networks and limit the population's access to health facilities, leading to a temporary decrease in reported cases, even if actual transmission remains stable or increases.

While our study identifies a macro-scale non-linear relationship between rainfall and temperature and malaria incidence, it is necessary to acknowledge that the study's spatial resolution of rainfall and temperature (0.10 degrees, ~11 km) averages out crucial micro-scale features that ultimately govern the formation of viable *Anopheles* breeding sites [59].

In that sense, the ERA5 data, while powerful for regional and municipal climate analysis, do not differentiate between microclimate characteristics, treating an entire grid cell as a homogenous unit. This scale mismatch is a well-documented challenge in environmental epidemiology for vector-borne diseases, as the processes that create larval habitats occur at a much finer scale than most satellite-derived environmental data can resolve [60].

Additionally, soil type is a key determinant of water retention and pooling. Clay-rich and loamy soils, common in many tropical regions, have low permeability and high water-holding capacity, which can create stable, long-lasting sunlit pools that are ideal for larval development, even after moderate rainfall events [61]. In contrast, sandy soils with high infiltration rates may not support the persistence of such habitats.

Future research could significantly enhance predictive accuracy by integrating very-high-resolution data, such as that from drone-based remote sensing or direct soil composition maps, to better characterize the microclimate and edaphic heterogeneity that facilitate vector proliferation at the local level [62].

The necessary exclusion of certain variables (e.g., MPI) was made to preserve the integrity of our causal inference and to prevent collider bias [43]. It is important to note that including multidimensional poverty of municipalities in the DAG position it as a collider. This is because Colombian regions with unhealthy climates (e.g., heavy rainfall, high temperature, permanent humidity, in sylvatic regions) have been historically neglected, leading to higher multidimensional poverty levels [16].

Therefore, adjusting for MPI in our model would have introduced collider bias, potentially distorting the estimated effect of rainfall and temperature on malaria incidence. These considerations highlight the delicate balance between including all potentially relevant variables related to malaria incidence (For example, estimating the change in the effect of rainfall and temperature due to the multidimensional poverty of municipalities) and maintaining the validity of causal inference in observational studies. Future research could explore alternative approaches to incorporate these complex relationships without compromising causal validity [63].

Although our TMLE approach incorporated municipality-level descriptors as effect modifiers in our DAG, we did not explicitly model the intra-municipality correlation structure through multilevel modeling, because, to our knowledge, currently, there is no software available with the capability of estimating the exposure-response curve with a causal machine learning approach, incorporating a hierarchical data structure. Future research should implement hierarchical modeling approaches that explicitly recognize the clustered nature of spatial-temporal health data, potentially incorporating random municipal effects to capture better the autocorrelation present in longitudinal ecological studies of climate-sensitive diseases.

Finally, the identified climatic thresholds provide actionable targets for operational integration into Colombia's malaria early warning systems. Specifically, municipalities experiencing temperatures between 20–25 °C combined with rainfall below 37 mm should trigger enhanced surveillance protocols and pre-positioning of rapid diagnostic tests and antimalarial drugs. Additionally, vector control interventions could be optimally scheduled during periods approaching these threshold conditions, maximizing cost-effectiveness by targeting high-transmission windows. Furthermore, exposure-response curves enable climate adaptation planning through seasonal forecasting models that incorporate climate variability associated with El Niño and La Niña phenomena, allowing health authorities to anticipate malaria outbreaks and allocate resources accordingly, thereby strengthening health security infrastructure in endemic regions.

## 4.4 Climate change implications and altitudinal range expansion

The temperature and rainfall thresholds identified in this study provide a critical baseline for understanding how climate change may reshape malaria transmission dynamics in Colombia. Global surface temperatures have increased by approximately 1.1 °C since pre-industrial times, with projections indicating further warming of 1.5-2.0 °C by mid-century under current emission trajectories [64]. This warming trend is expected to shift the geographic and altitudinal suitability for malaria vectors, potentially expanding transmission into previously unsuitable highland regions [65]. Empirical evidence from East African highlands has documented upward shifts in *Anopheles* vector distributions at rates of 6.5 meters per year, concurrent with warming trends of 0.5 °C per decade [66]. In Colombia specifically, modeling studies suggest

that climate change could increase the population at risk for malaria by 2–8% by 2050, with the most significant expansions occurring in mid-altitude municipalities (1,000–1,600 masl) currently at the margin of transmission suitability [11].

Furthermore, alterations in precipitation patterns associated with climate change—including increased frequency of extreme rainfall events and prolonged dry periods—may modify the established rainfall threshold we identified, potentially creating novel transmission windows in regions where seasonal patterns have historically limited vector breeding [67]. The non-linear exposure-response curves estimated in our study offer a methodological framework for projecting how these climate-driven changes will affect malaria incidence, enabling the development of adaptive early-warning systems that account for shifting climatic envelopes [68]. Future research should integrate these empirically-derived thresholds with downscaled climate projections to identify municipalities at highest risk for climate-induced malaria emergence, particularly those approaching the 25 °C temperature threshold or experiencing altered rainfall regimes that favor vector proliferation [69].

## 5 Conclusions

Our research, which implements causal machine learning at a municipal scale in Colombia, suggests that monthly temperature values between 15 and 25 °C favor the incidence of malaria, while temperatures above 25 °C reduce the occurrence of new cases. Similarly, monthly rainfall values below approximately 37 mm increase the incidence of malaria, whereas rainfall above this threshold results in a decrease in malaria incidence.

The suggested exposure-response curves for rainfall and temperature provide actionable intelligence for Colombia's malaria elimination program. Our estimates support targeted deployment of environmental monitoring systems in high-risk locations, where rainfall and temperature thresholds could trigger early warning alerts. Future research should consider additional key factors, such as health services data, to understand the role of socioeconomic potential sources of bias, while exploring sensor networks that capture microhabitat conditions at finer scales than current remote sensing technology allows. This study offers a framework for causal inference applications in vector-borne disease ecology, emphasizing the need to implement machine learning flexibility with rigorous causal assumptions testing.

## Supporting information

**S1 Fig. Records with spatial thinning to 25 km2 for the Anopheles species included in the analysis of vector co-occurrence.**
(DOCX)

**S1 Table. The table shows the episodes according to each climate agency and the consensus.** The rows without data in the column Consensus mean that in that month, there is no consensus about the episode occurring. The climate agencies included were the National Oceanic and Atmospheric Administration (NOAA), the Tokyo Climate Center (TCC), the Meteorological Office of the Australian Government (MOAG), and the Institute of Hydrology, Meteorology and Environmental Studies of Colombia (IDEAM).
(XLSX)

**S1 Text. Technical details of the causal inference analysis, machine learning implementation, and robustness and sensitivity tests.**
(DOCX)

## Acknowledgments

We acknowledge the Colombian Ministry of Health for access to information on malaria cases. We also express our gratitude to Mariano Altamiranda and Julián Ávila for their assistance with the methods for assessing the potential occurrence of malaria vectors.

## Author contributions

**Conceptualization:** Juan David Gutiérrez.

**Data curation:** Juan David Gutiérrez.

**Formal analysis:** Juan David Gutiérrez.

**Funding acquisition:** Juan David Gutiérrez.

**Investigation:** Juan David Gutiérrez.

**Methodology:** Juan David Gutiérrez.

**Project administration:** Juan David Gutiérrez.

**Resources:** Juan David Gutiérrez.

**Software:** Juan David Gutiérrez.

**Supervision:** Juan David Gutiérrez.

**Validation:** Juan David Gutiérrez.

**Visualization:** Juan David Gutiérrez.

**Writing – original draft:** Juan David Gutiérrez.

**Writing – review & editing:** Juan David Gutiérrez.

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
