## [Decision Letter · Decision Letter 0]

12 Nov 2025

PGPH-D-25-02542

Integrating Causal Inference and Machine Learning to Quantify Climate-Malaria Relationships: Evidence of Temperature and Rainfall Thresholds from Colombian Municipalities

Dear Dr. Gutiérrez,

Thank you for submitting your manuscript to PLOS Global Public Health. After careful consideration, we feel that it has merit but does not fully meet PLOS Global Public Health’s publication criteria as it currently stands. Therefore, we invite you to submit a revised version of the manuscript that addresses the points raised during the review process.

We look forward to receiving your revised manuscript.

Kind regards,

Kate Zinszer

Academic Editor

Journal Requirements:

1. Please send a completed 'Competing Interests' statement, including any COIs declared by your co-authors. If you have no competing interests to declare, please state "The authors have declared that no competing interests exist". Otherwise please declare all competing interests beginning with the statement "I have read the journal's policy and the authors of this manuscript have the following competing interests:"

2. Some material included in your submission may be copyrighted. According to PLOS’s copyright policy, authors who use figures or other material (e.g., graphics, clipart, maps) from another author or copyright holder must demonstrate or obtain permission to publish this material under the Creative Commons Attribution 4.0 International (CC BY 4.0) License used by PLOS journals. Please closely review the details of PLOS’s copyright requirements here: PLOS Licenses and Copyright. If you need to request permissions from a copyright holder, you may use PLOS's Copyright Content Permission form.

Potential Copyright Issues:

Figure 2 and Supplement 1: please (a) provide a direct link to the base layer of the map (i.e., the country or region border shape) and ensure this is also included in the figure legend; and (b) provide a link to the terms of use / license information for the base layer image or shapefile. We cannot publish proprietary or copyrighted maps (e.g. Google Maps, Mapquest) and the terms of use for your map base layer must be compatible with our CC-BY 4.0 license.

Reviewers' comments:

Reviewer's Responses to Questions

**Comments to the Author**

1. Does this manuscript meet PLOS Global Public Health’s publication criteria?

Reviewer #1: Yes

Reviewer #2: Partly

2. Has the statistical analysis been performed appropriately and rigorously?

Reviewer #1: Yes

Reviewer #2: No

3. Have the authors made all data underlying the findings in their manuscript fully available (please refer to the Data Availability Statement at the start of the manuscript PDF file)?

Reviewer #1: Yes

Reviewer #2: Yes

4. Is the manuscript presented in an intelligible fashion and written in standard English?

Reviewer #1: Yes

Reviewer #2: Yes

Reviewer #1: - I recommend writing '1600 meters' instead of '1600m'.

- The author fails to motivate the paper. Why is it worth it, the research Quentin? How can it be useful for public health policy in Colombia?

- Before estimating the causal model (temperature vs. malaria), I recommend including a spatial and temporal permutation model to localize the cluster. Neighboring municipalities share common temperatures and thus malaria exposure.

- The authors could include an isotherm and isometric map of Colombia to better illustrate the geographical distribution of the results.

- The link for the replication codes should be at the end of the document, not in the methods section.

- The discussion section is missing. How do these results compare to previous literature? How does the paper fill the gap in existing research?

Reviewer #2: The manuscript titled “Integrating Causal Inference and Machine Learning to Quantify Climate–Malaria Relationships: Evidence of Temperature and Rainfall Thresholds from Colombian Municipalities” presents an ambitious and technically sophisticated ecological study that combines causal inference methods with machine learning to estimate the effects of temperature and rainfall on malaria incidence in Colombia. The study’s scope aligns with the objectives of PLOS Global Public Health, particularly its emphasis on methodological innovation and data-driven public health insights for climate-sensitive diseases.

Overall Assessment:

The study is methodologically rigorous, conceptually strong, and addresses an important global health question linking climate variability to malaria transmission dynamics. The combination of Directed Acyclic Graphs (DAGs) with Targeted Maximum Likelihood Estimation (TMLE) represents a cutting-edge analytical framework rarely applied in environmental epidemiology. The dataset is extensive, covering 969 municipalities over 17 years, which provides substantial statistical power for causal modeling. The conclusions drawn are largely consistent with the data presented and contribute meaningfully to understanding threshold-based climate–malaria relationships.

1. Scientific and Methodological Rigor

The research design is robust and transparent. The use of DAGs to guide variable selection and control for confounding is methodologically sound, and the TMLE framework is appropriately chosen to estimate causal effects in complex observational datasets. The paper demonstrates thoughtful consideration of confounding, model specification, and sensitivity analyses through E-values and negative control tests.

However, a few points merit clarification:

• The exclusion of certain variables (e.g., MPI) to prevent collider bias is justified but could be more explicitly linked to the DAG interpretation for readers unfamiliar with causal reasoning.

• The negative control findings indicate residual confounding bias (p < 0.05). While transparently reported, the implications of this residual bias for causal interpretation should be discussed in greater depth.

• The assumptions of data independence and the decision not to model municipality-level clustering deserve further methodological justification, as repeated measures within municipalities likely violate independence assumptions.

Overall, the analytical framework is technically strong, but the authors should provide a clearer explanation of how these methodological limitations might influence the robustness of their causal estimates.

2. Statistical Analysis

The statistical analysis is advanced and implemented with rigor. The TMLE - gradient boosting integration, exposure-response curve estimation, and E-value sensitivity analyses demonstrate high analytical competency. The approach is well-suited for modeling non-linear relationships and estimating causal effects from high-dimensional data.

Nevertheless, the manuscript could benefit from:

• Simplified descriptions of the statistical models to improve accessibility for readers who may not be familiar with causal machine learning.

• Clarification on hyperparameter tuning procedures (e.g., justification for 1,500 estimators and learning rate selection) and whether cross-validation or out-of-sample validation was applied to mitigate overfitting.

• A clearer presentation of uncertainty intervals and their interpretation in the context of causal inference rather than traditional regression.

Despite these areas for improvement, the statistical analysis is appropriately conducted and sufficiently rigorous to support the stated conclusions.

3. Ethical and Publication Standards

The authors have stated that the study was approved by the Universidad de Santander Bioethics Committee and that only anonymized, aggregated surveillance data were used. There are no apparent ethical concerns regarding participant privacy or data integrity. The study adheres to STROBE reporting standards, which strengthens transparency and replicability. I found no evidence of dual publication or plagiarism.

All data sources and code repositories are clearly cited and publicly accessible, which aligns with PLOS’s open science policies.

4. Presentation and Language Quality

The manuscript is written in standard scientific English and is generally intelligible. However, it is overly dense and highly technical, particularly in the Methods and Results sections. Long sentences, heavy use of jargon, and algorithmic detail (e.g., references to “verbose outputs” and “gradient boosting hyperparameters”) make some sections difficult to follow for the journal’s broader public health audience.

The manuscript would benefit from:

• Language editing to improve readability, sentence flow, and paragraph structure.

• Condensation of repetitive methodological details (e.g., repeated mention of DAGs, confounding, and TMLE framework).

• Improved figure legends and summaries that allow independent interpretation of graphs and exposure-response curves.

Despite these stylistic issues, the text remains clear enough for reviewers to evaluate the science, and the English usage meets baseline journal standards.

5. Interpretation and Public Health Relevance

The findings that malaria incidence peaks at approximately 25 °C and 37 mm monthly rainfall are biologically plausible and consistent with previous literature. However, the discussion is heavily methodological and could better emphasize the public health significance of these results. The authors should explicitly describe how these climatic thresholds could be integrated into malaria early warning systems, inform vector control scheduling, or support climate adaptation planning in malaria-endemic regions.

Connecting the technical results more directly to applied health security and surveillance planning would significantly enhance the paper’s impact.

6. Recommendations for Improvement

1. Simplify technical sections to enhance accessibility to multidisciplinary readers.

2. Expand discussion on the implications of residual confounding and independence assumption violations.

3. Provide clearer linkage between climatic thresholds and practical malaria control strategies.

4. Condense repetitive text and streamline figures for clarity.

5. Consider professional language and structural editing before publication.

Final Evaluation

• Scientific soundness: Strong

• Methodological rigor: High

• Statistical validity: Appropriate, though slightly overtechnical

• Ethical standards: Fully met

• Readability: Moderate; needs editing for clarity

• Suitability for publication in PLOS Global Public Health: Yes, pending minor to moderate revision for clarity and accessibility.

**Do you want your identity to be public for this peer review?** For information about this choice, including consent withdrawal, please see our Privacy Policy

Reviewer #1: No

Reviewer #2: **Yes:** Oladayo David Awoyale

---

## [Decision Letter · Decision Letter 1]

6 Jan 2026

PGPH-D-25-02542R1

Integrating Causal Inference and Machine Learning to Quantify Climate-Malaria Relationships: Evidence of Temperature and Rainfall Thresholds from Colombian Municipalities

Dear Dr. Gutiérrez,

Thank you for submitting your manuscript to PLOS Global Public Health. After careful consideration, we feel that it has merit but does not fully meet PLOS Global Public Health’s publication criteria as it currently stands. Therefore, we invite you to submit a revised version of the manuscript that addresses the points raised during the review process.

We look forward to receiving your revised manuscript.

Kind regards,

Kate Zinszer

Academic Editor

Journal Requirements:

Reviewers' comments:

Reviewer's Responses to Questions

**Comments to the Author**

Reviewer #2: All comments have been addressed

Reviewer #3: (No Response)

publication criteria?

Reviewer #2: Yes

Reviewer #3: Yes

3. Has the statistical analysis been performed appropriately and rigorously?

Reviewer #2: Yes

Reviewer #3: Yes

4. Have the authors made all data underlying the findings in their manuscript fully available (please refer to the Data Availability Statement at the start of the manuscript PDF file)?

Reviewer #2: Yes

Reviewer #3: Yes

5. Is the manuscript presented in an intelligible fashion and written in standard English?

Reviewer #2: Yes

Reviewer #3: Yes

Reviewer #2: (No Response)

Reviewer #3: This manuscript presents a compelling approach to estimating the effects of rainfall and temperature on malaria incidence. The introduction is concise and effectively establishes the context. The methods and analyses appear robust and are clearly explained. I particularly appreciate the comprehensive explanation provided for each analysis. The results are succinct and well-articulated. While the discussion is somewhat lengthy and addresses numerous points, all are valid and interesting. I only have a few comments and suggestions for the author:

- Methods: On lines 111-112, please provide the reference for the STROBE guidelines utilized.

- Methods: Lines 120-121, regarding the sentence ‘Furthermore, data from municipalities exceeding 1,600 meters in elevation were excluded, based on the National Health Institute's established cutoff for malaria transmission viability [14]’. Is this reference accurate? Did the National Health Institute establish this cutoff? Please clarify whether the National Health Institute is the appropriate source, or if another publication should be cited for this cutoff.

- Discussion: I recommend including subheadings or sections within the discussion, as there are several important topics to address: the main results (temperature and rainfall gradients), the significance of using local data and models (a major strength of the paper), the methodological approach (machine learning, causal inference, etc.), and the limitations and challenges. Organizing the discussion in this way would enhance readability. This is a suggestion for the author, if it aligns with the journal's guidelines.

- Discussion: As an additional suggestion, given the length of the discussion, I believe it would be valuable to elaborate on the relevance of these results and methods in the context of climate change. Although the primary focus of the paper is not climate change, including one or two sentences on how this research can inform climate change studies would be beneficial. The Intergovernmental Panel on Climate Change is currently preparing its next Assessment Report and is seeking relevant research, particularly studies incorporating local data from the Global South, such as this one. Furthermore, the discussion could be strengthened by addressing how climate change may influence the values obtained, especially considering that some studies suggest mosquitoes are expanding their altitudinal range as average temperatures rise. I encourage the author to comment on this aspect.

**Do you want your identity to be public for this peer review?** For information about this choice, including consent withdrawal, please see our Privacy Policy

Reviewer #2: **Yes:** Oladayo David Awoyale

Reviewer #3: No

---

## [Editor Report · Decision Letter 2]

18 Jan 2026

Integrating Causal Inference and Machine Learning to Quantify Climate-Malaria Relationships: Evidence of Temperature and Rainfall Thresholds from Colombian Municipalities

PGPH-D-25-02542R2

Dear Mr. Gutiérrez,

We are pleased to inform you that your manuscript 'Integrating Causal Inference and Machine Learning to Quantify Climate-Malaria Relationships: Evidence of Temperature and Rainfall Thresholds from Colombian Municipalities' has been provisionally accepted for publication in PLOS Global Public Health.

Best regards,

Kate Zinszer

Academic Editor